# Breaking the crosstalk of the Cellular Tumorigenic Network by low-dose combination therapy in lung cancer patient-derived xenografts

Dennis Gürgen[1], Theresia Conrad[1], Michael Becker[1], Susanne Sebens [2], Christoph Röcken [3], Jens Hoffmann [1] & Stefan Langhammer [4✉]

Non-small cell lung cancer (NSCLC) is commonly diagnosed at advanced stages limiting treatment options. Although, targeted therapy has become integral part of NSCLC treatment therapies often fail to improve patient's prognosis. Based on previously published criteria for selecting drug combinations for overcoming resistances, NSCLC patient-derived xenograft (PDX) tumors were treated with a low dose combination of cabozantinib, afatinib, plerixafor and etoricoxib. All PDX tumors treated, including highly therapy-resistant adeno- and squamous cell carcinomas without targetable oncogenic mutations, were completely suppressed by this drug regimen, leading to an ORR of 81% and a CBR of 100%. The application and safety profile of this low dose therapy regimen was well manageable in the pre-clinical settings. Overall, this study provides evidence of a relationship between active paracrine signaling pathways of the Cellular Tumorigenic Network, which can be effectively targeted by a low-dose multimodal therapy to overcome therapy resistance and improve prognosis of NSCLC.

[1] EPO Experimental Pharmacology & Oncology, Berlin, Germany. [2] Institute for Tumorbiology, University of Kiel, Kiel, Germany. [3] Institute for Pathology, University of Kiel, Kiel, Germany. [4] life science consulting, Burgwedel, Germany. ✉email: langhammer@ls-consultant.net

Lung cancer is the leading cause of cancer-related mortality worldwide. Most cases of non-small cell lung cancer (NSCLC) are diagnosed at advanced metastasized stages and therefore indicated for palliative treatment. These mainly non-resectable carcinomas are treated with platinum-based chemotherapy alone or in combination with immune-checkpoint inhibitors, which are the mainstay regimens in the absence of predictive, targetable oncogenic mutations and an expression of the immune-checkpoint inhibitor Programmed death ligand-1 (PD-L1) of <50%[1]. In patients with NSCLC of squamous and non-squamous histology bearing a PD-L1 expression >50% an improved efficacy for a monotherapy with pembrolizumab compared to a platinum-based chemotherapy was observed and thus became the standard of care for these patients. For the treatment of NSCLC patients characterized by targetable and predictive oncogenic mutations, many different targeted therapies have already been approved or are under development, including drugs targeting Epidermal Growth Factor Receptor mutations (EGFR+), Anaplastic Lymphoma Kinase (ALK)/proto-oncogene tyrosine-protein kinase (ALK/ROS+) fusions, proto-oncogenes KRAS and B-Raf (BRAF+) mutations, respectively, as well as neurotrophic tyrosine kinase receptor (NTRK+) mutations[2–4].

The 5-year survival rate of NSCLC patients is still below 25% and there is still a pressing medical need for the development of effective treatment options[5]. NSCLC patients without targetable oncogenic mutations and without PD-L1 expression are still a major challenge and there is a need for experimental generation of new, rational combination strategies with targeted therapies addressing factors of the Cellular Tumorigenic Network[6,7], as apart from the tumor cells, non-neoplastic cells such as cancer-associated fibroblasts, endothelial cells, and immune cells contribute to tumor growth. Tumor cell signaling reprograms these cell types inducing processes such as angiogenesis, inhibition of apoptosis, immune evasion, and synthesis of soluble tumor microenvironment components. During the evolving tumor development, signaling networks within the tumor are reshaped further and programs such as hypoxia led to heterogeneity within specific tumor areas. This interdependence could be the initial cause of *de novo* and adaptive drug resistances in many cases[8–10].

A simultaneous and distinct targeting of signaling networks is intended to break the intercellular crosstalk and to overcome *de novo* and prevent adaptive drug resistances. Noteworthy, the direct suppression of cellular interdependency is most likely be achieved by targeting critical paracrine signaling axes of cell surface receptors and their respective ligands. These pathways are blocked at the receptor level while inhibiting multiple generic intracellularly connected downstream pathways involved in tumor development and treatment resistance such as the mitogen-activated protein kinase (MAPK), Janus kinase and the signal transducer and activator of transcription protein (JAK/STAT), phosphoinositide 3-kinase (PI3K), protein kinase B (AKT), and protein kinase C (PKC) pathways (Fig. 1)[11].

In order to break the crosstalk within the tumorigenic network in late-stage NSCLC patients as described above, previously, we postulated five criteria for selecting a combinational targeted therapy approach[7].

(i)   Target expression or pathway activation proven by expression and/or mutational analysis
(ii)  Target should be part of a paracrine signaling pathway mediating intercellular interdependency within the Cellular Tumorigenic Network
(iii) Drug combination should target non-overlapping intercellular signaling axes of the Cellular Tumorigenic Network
(iv)  Selected drugs should have shown proven activity in clinical studies

(v)   Manageable safety profile of drug combinations.

In alignment with these criteria, we have chosen a low dose combination therapy with cabozantinib, afatinib, etoricoxib, and plerixafor addressing each cell type involved in the paracrine Cellular Tumorigenic Network (Table 1). The intention of this drug regimen is to disrupt paracrine substitutional signaling within the tumorigenic network in order to prevent adaptive drug resistances: The tumor-promoting effects of endothelial cells, (i) angiogenesis, (ii) tumor cell survival, and (iii) metastasis, are targeted by inhibition of the VEGF-VEGFR2 axis with cabozantinib. The interrupted VEGF-VEGFR2 signaling is substituted by cancer-associated fibroblasts and tumor cells through the secretion of HGF, EGF, and SDF-1α, by inducing the expression of their corresponding receptors and by direct cell to cell interactions[12–15]. Therefore, this effect is intended to be counter-regulated using afatinib for targeting the EGF-EGFR axis and plerixafor for targeting the SDF-1α—CXCR4 axis in the chosen therapy regimen. In addition, cabozantinib also targets the HGF-MET axis. The interruption of EGFR signaling by afatinib would lead to substitution by cancer-associated fibroblasts and tumor cells secreting HGF and SDF-1α[14,16,17]. SDF-1α secreting cells such as cancer-associated fibroblasts and tumor endothelial cells may substitute for interrupted EGFR signaling by paracrine signaling via CXCR4[18]. Again, these anticipated resistances will be prevented by applying the inhibitors of the paracrine axes of HGF-MET and SDF-1α - CXCR4 through cabozantinib and plerixafor in this therapy regimen. A potential resistance against VEGFR inhibition with cabozantinib may also be initiated by the production of PGE2 via COX2[19]. This effect is intended to be counter-regulated using etoricoxib in this therapy regimen (Fig. 1)[7].

## Results

**Evidence for a relationship between gene expression and therapy resistances in patient-derived NSCLC tumors.** In order to assess the five criteria postulated for overcoming drug resistances 38 well characterized patient-derived, human NSCLC tumor xenograft models (PDX) from the EPO PDX panel were analyzed for mRNA expression of (i) paracrine signaling pathways mediating intercellular interdependency within the Cellular Tumorigenic Network that were (ii) non-overlapping and have been described (iii) as relevant for tumor proliferation previously and druggable by approved inhibitors[6]. Specifically, relative mRNA expression levels of *EGFR, EGF, VEGFR2 (KDR), VEGFA, CXCR4, SDF-1α (CXCL12), MET, HGF,* and *COX2* were analyzed in this sample set and all genes were found to be expressed at different transcript levels.

First, the specific marker for squamous cell carcinoma *TP63*[20] was analyzed as control of the biological validity of this data set and correlated with the known tumor histology of these tumors (Fig. 2a). Twenty-six of these tumors were characterized as squamous cell carcinomas and 12 tumors as adenocarcinomas according to their histopathological classification. In 23 out of 26 squamous NSCLC tumors, high *TP63* expression was detected (expression value > 12). In contrast, the analysis of *TP63* transcripts in adenocarcinomas indicated very low abundance or absence of detection in this NSCLC subpopulation. Therefore, *TP63* could be used as a valid marker for squamous cell carcinoma histology within this data set. In conclusion, the high rates of sensitivity (0.88, 95% confidence interval (CI): [0.70; 0.98]) and specificity (1.00, 95% CI: [0.74; 1.00]) of TP63 expression demonstrates the high biological relevance of our RNA sequence NSCLC PDX data set.

Utilization of ssGSVA in combination with a permutation-based test provided an indication for the importance of our

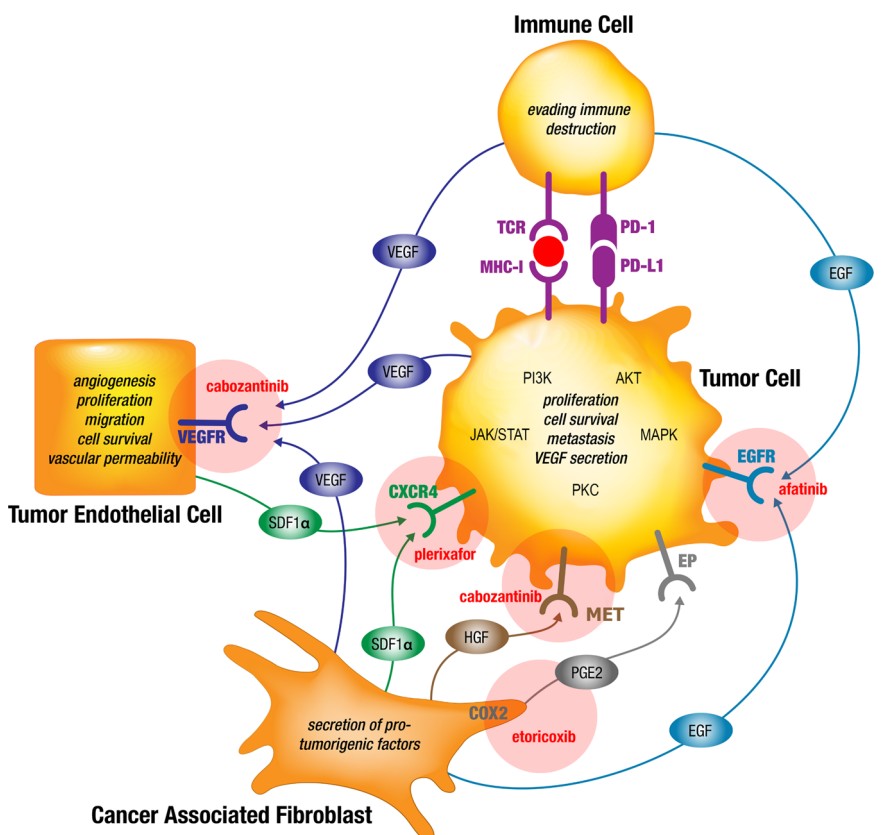

**Fig. 1 Simplified model of the chosen combination therapy regimen overcoming drug resistances by simultaneous targeting of interdependent signaling in the c Cellular Tumorigenic Network of NSCLC tumors using already marketed drugs.** Signaling axes of VEGF-VEGFR, EGF-EGFR, SDF-1-CXCR4, HGF-MET, COX2-PGE2-EP, and PD-1-PD-L1 are exemplary shown for some of the known paracrine pathways binding the Cellular Tumorigenic Network in NSCLC tumors. Evidence is provided that resistances to targeted therapy drugs are partly based on substitutions for inhibited pathways and on crosstalk of generic intracellularly connected downstream pathways involved in tumor development and treatment resistance such as MAPK, JAK/STAT, PI3K, AKT, and PKC[11]. Therefore, combined targeted therapies against selected pathways may overcome primary and secondary drug resistances. The combination therapy drugs cabozantinib, afatinib, plerixafor, and etoricoxib are provided adjacent to the respective targeted pathways (figure under Creative Commons Attribution 3.0 License updated from Langhammer and Scheerer, 2017[7]). PGE2, prostaglandin E2; VEGFR, vascular endothelial growth factor receptor; EGFR, epidermal growth factor receptor; COX2 (PTGS2), cyclooxygenase 2; HGF, hepatocyte growth factor; red circle: tumor neo-antigen.

**Table 1 Characteristics of the low dose combination drug regimen applied in this study compared to commonly used doses.**

| Target | Drug | Selected low dose combination therapy | | Common dosing in vivo models | |
|---|---|---|---|---|---|
| | | Dosing | Schedule | Dosing | Schedule |
| VEGFR2 (KDR)/MET | Cabozantinib | 15 mg/kg | 5 days on/2 days off for 4 cycles | 30–60 mg/kg[26,49] | once daily |
| CXCR4 | Plerixafor | 5 mg/kg | | 5 mg/kg[28] | |
| EGFRwt | Afatinib | 15 mg/kg | | 20 mg/kg | |
| COX2 (PTGS2) | Etoricoxib | 10 mg/kg | | 10 mg/kg[29] | |

selected nine genes within the tumorigenic network. The obtained high enrichment scores for all 38 NSCLC PDX models reveal an over-representation of the selected genes within a set of 19096 HGNC-annotated genes. The depicting chart of scaled enrichment scores distributes our 38 NSCLC PDX cohort into several segments representing the biological diversity and individual characteristics of each PDX model (Fig. 2b). The permutation-based test comparing the selected gene set with randomly generated gene sets shows the statistically significant over-representation of our selected nine genes for eight PDX models. Interestingly, a greater proportion of these models (five out of eight) were insensitive against a standard of care treatment based on their initial characterization in the EPO tumor biobank and therefore classified as highly resistant tumors with progressive

disease (PD) as response according to the Response Evaluation Criteria In Solid Tumors (RECIST) criteria[21] within the PDX panel.

In order to evaluate a potential correlation between gene expression and treatment responses, we analyzed the expression data of the 38 NSCLC PDX models comprising 19096 HGNC-annotated genes and subsequently performed hierarchical clustering (Supplementary Fig. 1). When magnifying for the nine selected genes of high importance in the tumorigenic network. The resulting heatmap divides our NSCLC PDX panel into two main clusters, which reflect very well the histology-based gene expression of adenocarcinoma and squamous cell carcinoma. Only four squamous models were found to cluster with the adenocarcinoma subset of 12 PDX models. (Fig. 2c). Interestingly,

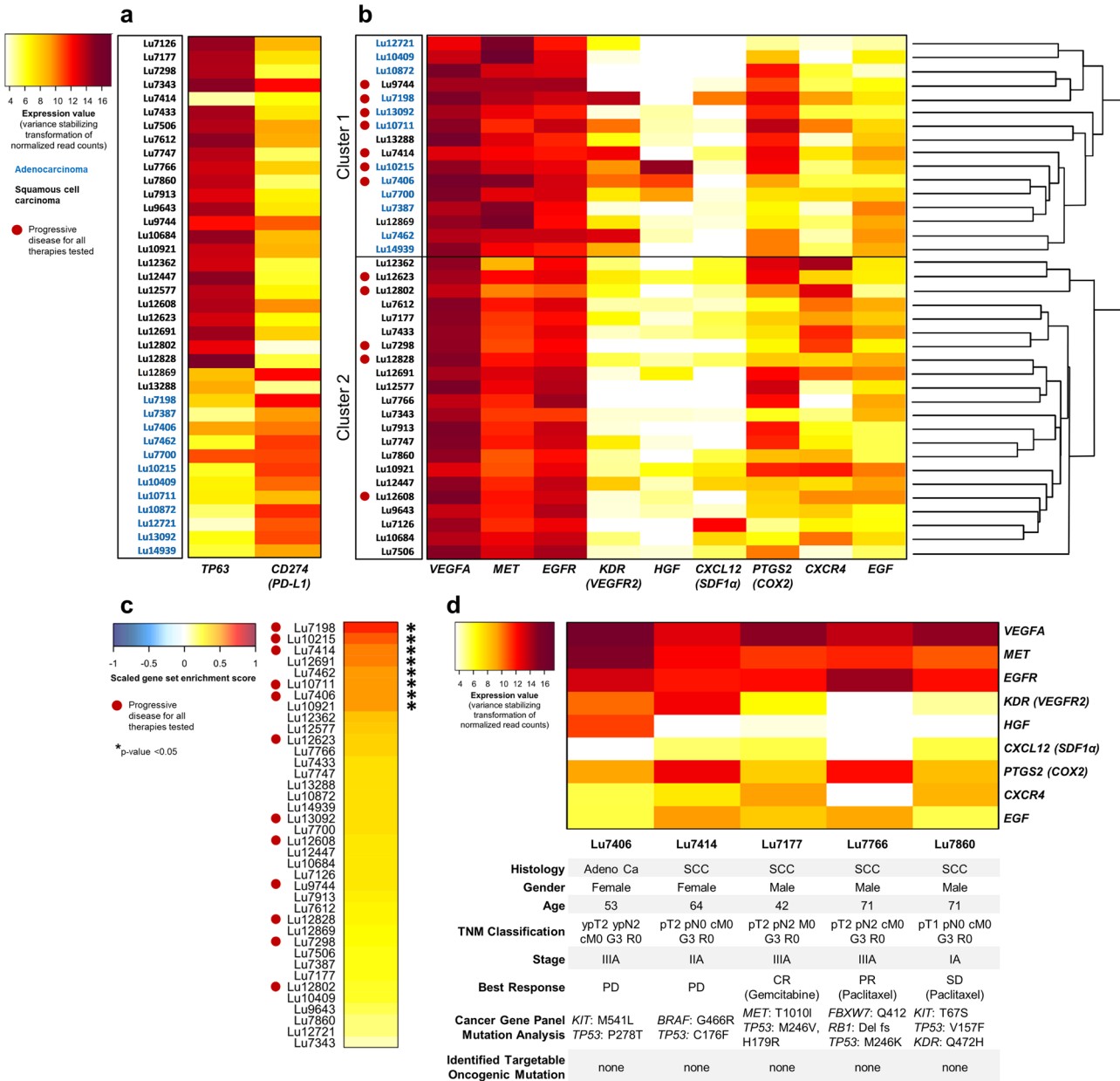

**Fig. 2 Characterization of the NSCLC PDX panel for target gene expression, single-sample GSVA and selected PDX models for targeting the Cellular Tumorigenic Network with the combination therapy regimen. a** *TP63* expression is a highly specific and selective marker for squamous cell carcinoma histology. The heatmap represents the relative expression of *TP63* and *PD-L1 (CD274)* in all 38 NSCLC PDX models from the EPO tumor biobank. 23 out of 26 NSCLC tumors with squamous cell carcinoma histology (as available in the respective histopathology report) show *TP63* expression values higher than the predefined cut-off 12. In contrast, the *TP63* expression level of all of the NSCLC tumors with adenocarcinoma histology is considerably lower than the cut-off. *PD-L1* expression is detected in the majority of tumors and seems to be higher expressed in the NSCLC tumors with adenocarcinoma histology. **b** Single-sample gene set variation analysis (ssGSVA) combined with a permutation-based test for the nine selected genes of the tumorigenic network. The high enrichment scores for all 38 NSCLC PDX models represent the over-representation of the nine selected genes within a set of 19096 HGNC-annotated genes. For eight PDX models, the selected gene set is statistically significantly over-represented compared to randomly generated gene sets. Thereof, five out of eight models were classified as highly resistant tumors within the PDX panel. **c** Relationship between gene expression and therapy resistances in NSCLC tumors. The heatmap represents the relative expression of *CXCR4, SDF-1α (CXCL12), EGF, EGFR, HGF, MET, VEGFA, VEGFR2 (KDR)*, and *COX2 (PTGS2)* in 38 NSCLC tumors from the EPO tumor biobank. The hierarchical clustering dendrogram is based on the expression data of 19096 HGNC-annotated genes. It shows the division of the analyzed PDX models into two main clusters, largely reflecting the histology-based gene expression of adenocarcinoma and squamous cell carcinoma. The complete dendrogram is provided in Supplementary Fig. 1. **d** Characterization of selected NSCLC tumors for treatment with the combination therapy regimen. Expression values, patient demographics, tumor characteristics, best response per RECIST in patient-derived xenografts, and identified mutations in the Illumina Cancer Gene Panel are shown. Only NSCLC tumors without targetable oncogenic mutation were selected including highly treatment-resistant tumors (Lu7406, Lu7414) and tumors with better RECIST responses and lower mRNA expressions of the target genes (Lu7177, Lu7766, and Lu7860). PDX tumors with progressive disease (PD) as the best response for all therapies tested previously are indicated with a red circle. Adeno Ca: adenocarcinoma; SCC: squamous cell carcinoma; Del fs: deletion frameshift; PD: progressive disease; SD: stable disease; PR: partial response; CR: complete response.

we found 44% highly resistant PDX models with progressive disease (PD) as a best response (7 out of 16) within Cluster 1. In contrast, only 23% of highly resistant tumors were found in Cluster 2 (5 out of 22).

This result indicates a relationship between the expression of specific genes and treatment resistances towards standard anti-tumor therapies including chemotherapies and targeted therapies. However, this finding does not allow the direct identification of particular genes causative for the observed clustering. Astonishingly, we found that all nine selected genes of the tumorigenic network were significantly expressed and that the seven highly resistant PDX models with progressive disease (PD) identified in Cluster 1 exhibit the highest transcript expression for *VEGFA, MET, EGFR, KDR*, and *PTGS2* within the complete NSCLC cohort.

Based on these results, five NSCLC PDX tumors without targetable oncogenic mutations were selected by means of their target gene expression and resistance profile, including highly resistant tumors with progressive disease (PD) (Lu7406 and Lu7414) and tumors with at least stable disease (SD) or even better responses (PR, CR) for any chemo or targeted monotherapy tested previously. The existence of targetable oncogenic mutations was excluded by utilizing the Illumina amplicon cancer panel (TSACP), sequencing 212 regions in 48 cancer-related genes including *EGFR, BRAF, KRAS, MET*, and *RET* (Fig. 2d). As an example, in this panel, the *KRAS G12C* mutation was identified in the highly treatment-resistant PDX tumor Lu7198 and therefore this model was excluded from further experiments at this stage. In none of the selected PDX models a rearrangement of *ALK, ROS, RET*, or *NTRK* was identified as assessed by RNASeq mutational analysis.

The selected NSCLC PDX tumors were either histologically categorized as adenocarcinoma (Lu7406) or squamous cell carcinoma (Lu7414, Lu7177, Lu7766, and Lu7860), which was confirmed by the *TP63* expression analysis with the exception of false-negative detection for Lu7414 as discussed above (Fig. 2a). All selected tumors from the NSCLC PDX panel were previously analyzed and reported for the identification of *LRP12* DNA methylation as a predictive biomarker for carboplatin resistance[22].

**Complete suppression of tumor growth and absence of toxicity in all NSCLC PDX models treated with the combination therapy regimen.** In total, 16 nude mice bearing one of the selected NSCLC PDX tumors were randomized for treatment with the combination therapy regimen consisting of low-dose cabozantinib, afatinib, plerixafor, and etoricoxib for at least four cycles with 5 days on and 2 days off treatment controlled by a placebo ($N = 18$) or monotherapy treatment ($N = 12$) group (Table 1). Additionally, the tolerability of our combination therapy regime was tested for a longer treatment cycle with daily application up to 8 days in the initial Lu7406 trial without adverse effects.

The safety profile of this combination regimen was assessed by the continuous measurement of body weight of treated mice compared to the placebo group as an indicator for toxicity. Throughout the experiments, only in the Lu7414 trial a body weight reduction between 5–13% was detected for the combination during the second cycle after day 4 resulting in drug holidays from day 5–7 in this cycle. However, the cumulative dose of these mice was the same compared to all other animals treated with the combination since the first cycle was tested with 6 days on and 1 day off treatment. In none of the remaining mice at any time point a difference in body weight between the verum and the placebo or the monotherapy group was observed. In general, all

mice included in these experiments either gained weight during the course of treatment or remained at the baseline body weight level with minimal variances (Fig. 3). None of the mice had to be discontinued from treatment, died, or had to be euthanized during the course of treatment. This observation reflects the beneficial safety profile of this combination regimen and is in alignment with the low-dose therapy compared to regular concentrations used of these drugs in in vivo experiments (Table 1).

16 out of 16 NSCLC PDX tumors combined from all models showed a treatment response for the applied combination regimen as evaluated per RECIST analysis (Fig. 4). In 13 tumors a partial response (PR) and in three tumors a stable disease (SD) was induced. In contrast, in 29 out of 30 placebo or monotherapy treated tumors progressive disease (PD) was observed. Only for a single tumor of the cabozantinib group a partial response was observed (Fig. 3b). Thus, the ORR is 81% and the CBR is 100% across all NSCLC PDX tumors treated with the combination regimen. In none of the mice treated with the combination regime a tumor outgrowth was observed during the treatment period. In contrast, all tumors in the placebo- or monotherapy groups showed a progressive disease on the last day of treatment. In addition, the growth of all tumors in the combination regimen group remained suppressed in the follow-up period for up to 18 days (Lu7860) (Fig. 3a).

Overall, these data indicate that combined targeting of the paracrine Cellular Tumorigenic Network elicits strong anti-tumoral effects in NSCLC PDX tumors which are characterized by profound resistances to standard therapies.

**PD-L1 expression in residual tumor specimens of the combination therapy group.** In order to assess the composition of the tumor and stromal compartment in the residual tumors treated with the combination regimen and respective placebo groups, tumors Lu7414, Lu7177, and Lu7766 were analyzed by immunohistology for pan-cytokeratin (tumor compartment), α-SMA (stromal myofibroblasts), and CD31 (blood vessels). No histological differences were observed between tumors of the treatment groups and tumors of the respective placebo groups (Fig. 5a). In addition, immunohistochemical staining of the immune checkpoint regulator PD-L1 was performed in order to assess the expression of an additional important anti-tumor target in NSCLC.

Even though no differences between treatment and placebo groups in the three tumor models could be observed, a considerable membranous expression of PD-L1 was detected in all residual tumors of either group at different levels. This result again reflects the heterogeneous expression levels of PD-L1 in these models (Fig. 2a) Importantly, the expression pattern of PD-L1 was heterogeneous across the tumor specimens, ranging from areas with weak expression up to areas with strong PD-L1 expression (Fig. 5b). In summary, these data indicate that the combination therapy does not impair PD-L1 expression providing the opportunity of treatment regimens in combination with immune-checkpoint inhibitors.

## Discussion
Despite some impactful advances in the treatment of NSCLC without targetable oncogenic mutations and without expression of PD-L1 resulting in an ORR of about 50%[1], the unmet medical need for many patients in this population remains high[5].

Our previously published hypothesis for the treatment of these NSCLC tumors by breaking the Cellular Tumorigenic Network has now been experimentally proven by the present study using PDX tumor models most closely resembling the clinical situation

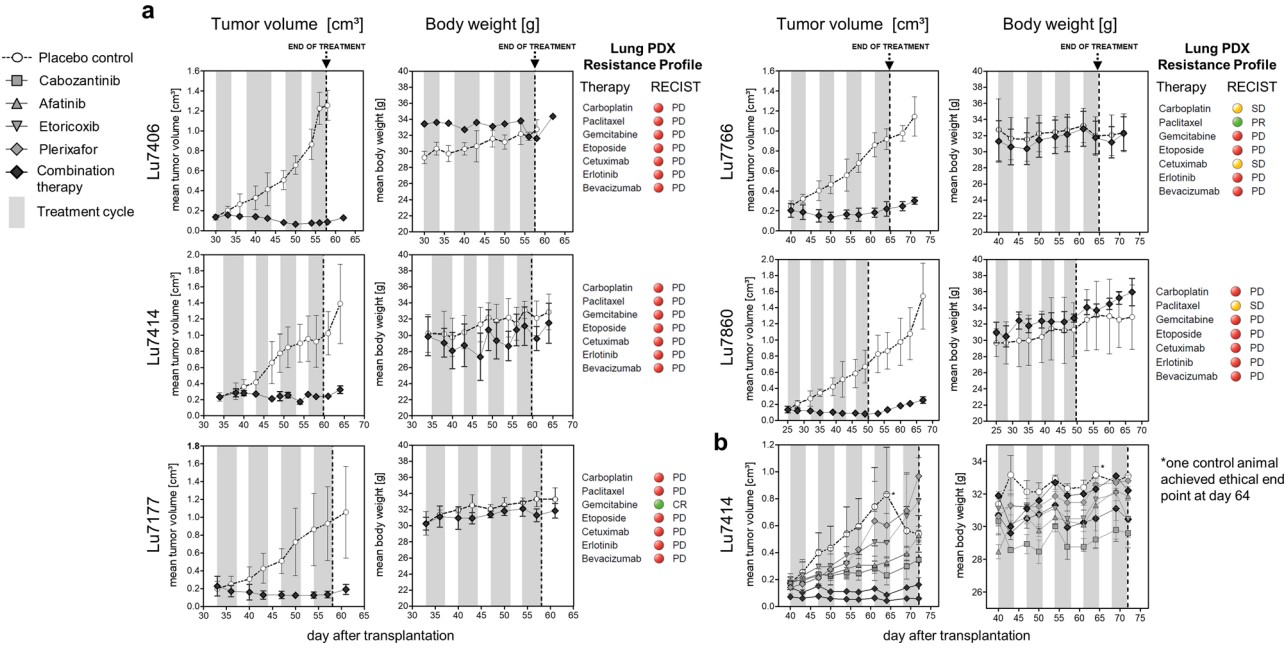

**Fig. 3 Efficacy of the combination therapy regimen in NSCLC PDX mouse models. a** In all trials, treatment with the combination therapy regimen (cabozantinib, plerixafor, afatinib, and etoricoxib) was highly effective in preventing tumor propagation and tumor out-growth achieving stable disease and partial response as best response criteria in all experiments. Stable body weight throughout the complete trials indicates a distinguished safety profile of the applied combination regimen. Therapy resistance profiles of lung PDX models per RECIST are indicated and visualized by the red circle for progressive disease (PD), yellow circle for stable disease (SD), and green circle for partial and complete response (PR, CR). All groups were conducted with $n = 3$ animals, except for the combination therapy of Lu7406 which was conducted as single a mouse trial ($n = 1$). **b** The experiment in Lu7414 with single compounds alone versus the combination therapy regimen reveals a superior effect of the low dose combination therapy. All groups were conducted with $n = 3$ animals, except for the combination therapy of Lu7414 which was conducted in a trial with two mice ($n = 2$). Error bars represent the standard deviation within each group with $n \geq 3$. End of treatment cycles are indicated by a vertical dotted line.

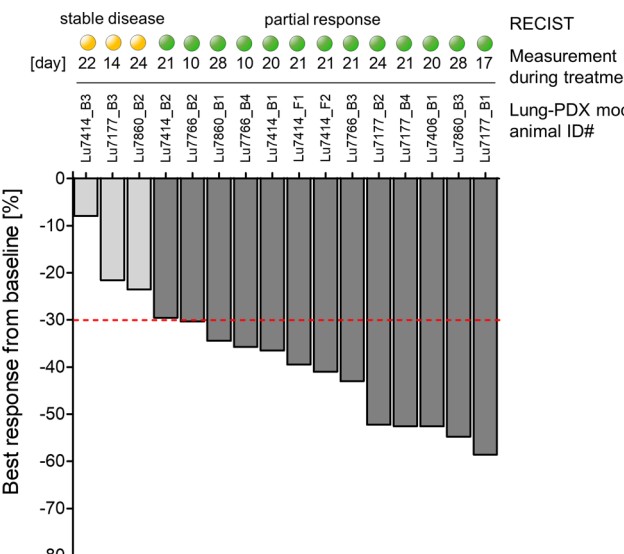

**Fig. 4 Best response per RECIST analysis in all NSCLC patient-derived xenograft tumors treated with the combination therapy regimen.** Depiction of all 16 NSCLC PDX tumors treated with the combination therapy as best response measured from baseline tumor size in percent. Response definition was calculated per RECIST criteria with partial response (PR, visualized by a green circle) defined as 30% reduction and stable disease (SD, visualized by a yellow circle) defined as neither response or progression in the sum of the longest tumor diameter.

and being highly predictive for treatment responses in cancer patients[23].

We provide evidence that the expression of a respective set of target genes is important for NSCLC development, homeostasis and progression. Our ssGSVA obtained enrichment scores indicate the relevance of the underlying biological function of the selected pathways in NSCLC disease and treatment. Observing a relatively low statistically significant over-representation of the nine target genes for only 8 of 38 models by permutation-based test is even more confirmative highlighting the great range of individual NSCLC characteristics and phenotypes rather than pointing out a bias with our analysis.

Furthermore, we showed that gene expression of a specific gene set of the tumorigenic network might be related to treatment resistance against chemo- or targeted monotherapies. A high expression of this nine gene set correlates mainly with disease progression in in vivo sensitivity testing with chemo- and targeted therapies.

All tumors randomly selected from the NSCLC PDX biobank panel responded well to this low-dose combination therapy despite showing different expression levels of the target genes with simultaneous coverage of diverse clinical phenotypes. Our finding of a potential relationship between the expression of target genes within the tumorigenic network and the response to different anti-tumor therapies should be further expanded verifying our investigations by utilizing an even more extensive data set of well characterized NSCLC tumors.

Nevertheless, to the best of our knowledge, for the first time, we show that the simultaneous and distinct targeting of paracrine signaling from different cell types of the Cellular Tumorigenic

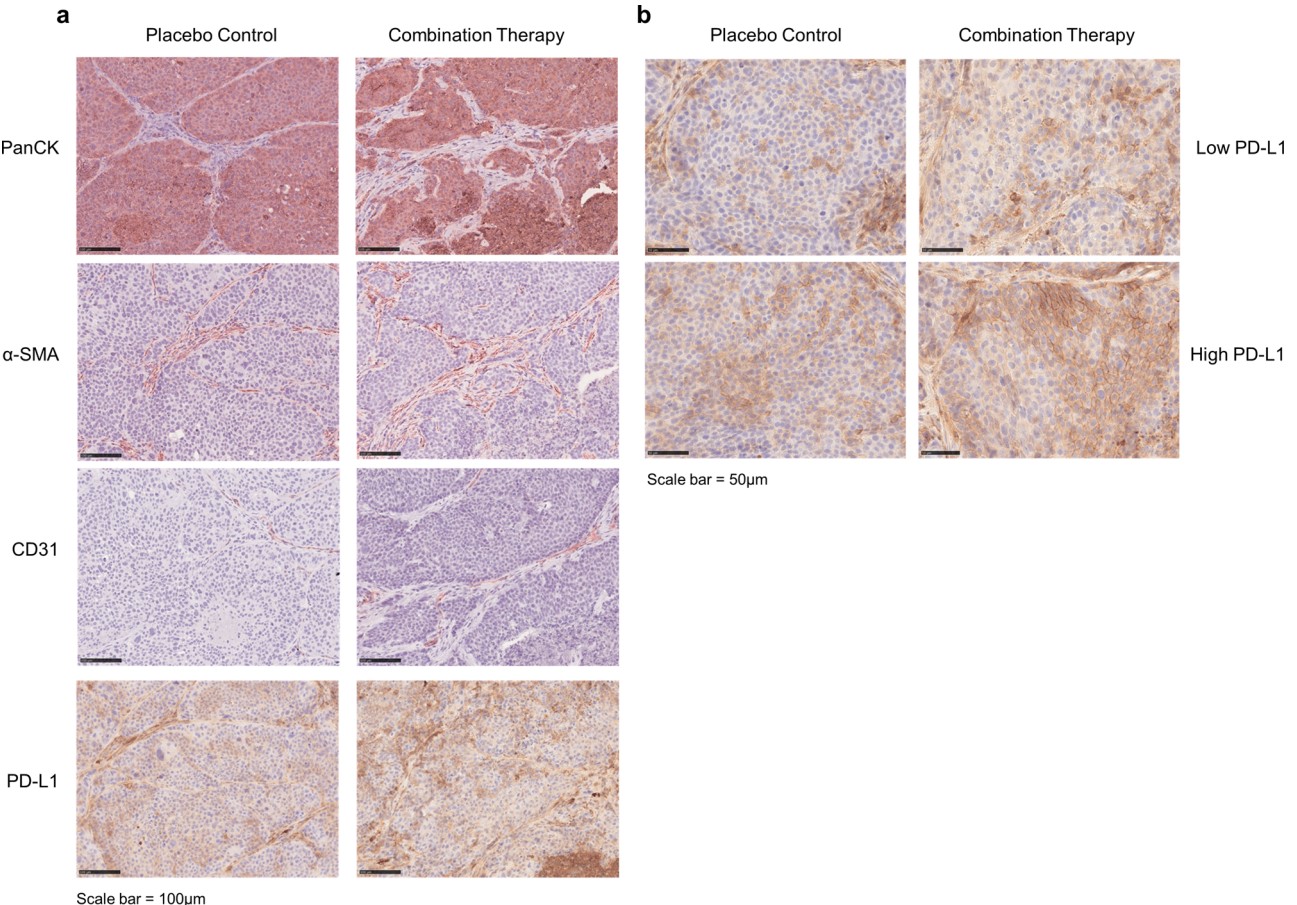

**Fig. 5 Immunohistochemical analysis of tumor structure and PD-L1. a** Representative pictures of immunohistochemical analysis of NSCLC PDX treated with the combination therapy and respective placebo controls for pan-cytokeratin (tumor compartment), α-SMA (stromal myofibroblasts), CD31 (blood vessels) and PD-L1, the latter stained as a potential additional therapeutic target. Scale bar = 100 μm. **b** Differentially treated PDX tumors exhibit different patterns of PD-L1 expression (low versus high PD-L1 expression). Scale bar = 50 μm.

Network breaks the intercellular crosstalk and is highly effective in tumors that are resistant to monotherapies of chemotherapeutics and targeted drugs against EGFR and VEGFR. In all treated NSCLC tumors without a targetable oncogenic mutation randomly selected from a panel of 38 NSCLC tumors, growth was completely suppressed by the applied low dose combination regimen. Among them were two highly treatment-resistant tumors that did not respond to any tested therapy previously (Lu7406 and Lu7414). During the complete treatment duration, no tumor outgrowth was observed in either PDX model, indicating a strong therapy efficacy by preventing development of an adaptive resistance against this regimen. After treatment discontinuation growth suppression remained up to 18 days (Lu7860) with only slight increase in tumor growth. This indicates a long-lasting and sustained effect of the combination therapy on tumor biology.

The combination therapy regimen that was compiled on the five criteria published previously[7] can be considered as a low dose treatment regimen. The two drugs with a more pronounced safety profile in this therapy regimen are cabozantinib and afatinib. Cabozantinib is approved for the treatment of renal cell carcinoma and hepatocellular carcinoma and is efficacious in different tumor models at dosages between 30–60 mg/kg once a day[24–26]. Therefore, the low dose drug combination used in our study only includes 50% or less of typically used cabozantinib concentrations in recently published preclinical studies. Afatinib is approved for the treatment of NSCLC either with activating EGFR mutation or in squamous cell carcinoma under or after progression on platinum-based chemotherapy and is efficacious in tumor models at dosages from 20 mg/kg or higher once a day[27]. In the combination therapy regimen used in this preclinical study, both drugs were used below these drug doses, each at 15 mg/kg and in addition to that paused after every 5 days of treatment. Plerixafor, approved for stem cell mobilization, and etoricoxib, approved for pain treatment and degenerative and auto-inflammatory diseases, were used at a commonly used concentration level in in vivo models[28,29]. The usage of this combination regimen showed an excellent safety profile and might lead to less pronounced side effects in patients. A database analysis for potential drug–drug interactions of this drug combination only revealed a potentially moderate clinically interaction between cabozantinib and afatinib that may result in a plasma level increase of afatinib when administered after exposure to a P-glycoprotein (P-gp) inhibitor. A monitoring of patients for the onset of adverse effects is recommended. Due to unavailability in this DDI database etoricoxib was replaced with celecoxib, another drug of the identical drug class of selective COX2 inhibitors[30].

Together, with its excellent efficacy, this combination would be superior to a prescribed medication of single cabozantinib or afatinib at higher drug dosages.

Treatment with the combination therapy did not seem to change the histopathological composition of tumors when compared to placebo treatment regarding the extent of the tumor and the stroma compartment as well as the number of blood vessels.

Since the studies have been performed in nude mice, it remains open whether this treatment regimen impacts the infiltration, composition, and effector phenotype of T cells. However, our studies also revealed that the expression of PD-L1, whose therapeutic targeting has essentially improved the prognoses of even metastasized NSCLC patients, was not affected by the combination therapy, offering the opportunity of combination therapies including immune-checkpoint inhibitors.

Taken together the simultaneous inhibition of pathways forming the Cellular Tumorigenic Network in NSCLC tumors can overcome resistance mechanisms of targeted therapy drugs. All of the drugs used in this combination regimen are approved for different indications, and therefore their clinical profiles such as pharmacokinetics, pharmacodynamics, and toxicities are well known. They are readily available as study drugs for usage in a clinical trial. In order to identify NSCLC patient subgroups that may respond to this therapeutic concept, inclusion criteria should require the proof of expression of the respective target gene set. Collected tumor specimens can easily be used for mRNA expression analysis by specific RT-real time PCR or by a customized gene expression array. The analysis of mRNA expression as a predictive biomarker is already in clinical development[31].

Based on our results we encourage the further validation of this combination therapy in additional cancer models and we recommend the setup of a clinical study in advanced stage NSCLC patients without a targetable oncogenic mutation combining the drugs cabozantinib, afatinib, plerixafor and etoricoxib in a low dose treatment regimen.

## Methods

**Whole transcriptome sequencing (RNASeq), data processing, and data analyses**. RNA sequencing was performed for 38 NSCLC models, comprising 12 adenocarcinomas and 26 squamous cell carcinomas. One sample each from the established patient-derived xenograft models was analyzed.

**RNASeq**. Total RNA was isolated from snap-frozen PDX tissue using Invitrogen™ TRIzol™ Reagent (Cat-No. 15596018, ThermoFisher Scientific, Germany). 50 mg of tumor tissue was disrupted in 1.5 ml TRIzol™ using a gentleMACS™ dissociator and M tubes (Cat-No. 130-093-235 & 130-096-335, Miltenyi Biotec, Germany). RNA integrity was evaluated with Agilent Bioanalyzer 2100 and RNA 6000 Nano Kit (Cat-No. G2939BA & 5067-1511, Agilent, Germany). RNASeq libraries were prepared using illumina® TrueSeq™ Stranded mRNA Library Prep Kit. The 100-bp-PE-sequencings ran on an illumina® HigSeq 2500 device with a depth of 80–100 Mio reads (40–50 Mio cluster) (Illumina, Cambridge, UK).

**Data processing**. The quality of reads was proven by the tool FastQC[32] version 0.11.8. For classification of the xenograft-derived sequence reads (human/mouse read splitting) the tool xenome[33] version 1.0.1 and human genome hg38 as graft reference as well as mouse genome mm10 as host reference were used. Mapping of human-specific reads against the *homo sapiens* reference hg38 was done using the STAR aligner[34] version 2.6.1a. Quality of mapping was proven by QualiMap[35] version 2.2.1 and quantification of transcripts was performed by eXpress[36] version 1.5.1.

**Expression data analysis**. The expression data analyses were performed in R[37] version 4.0.3 using packages provided by Bioconductor 3.7[38]. Package DESeq2[39] was used to normalize gene level raw read count data by division with size factors or normalization factors, calculate a variance stabilizing transformation and transform the normalized read count data. The transformed data are on the log2 scale for large counts. Gene names are utilized in accordance with the approved annotations of the HUGO Gene Nomenclature Committee (HGNC)[40]. Further analyses were based on the resulting expression values of 19096 HGNC-annotated genes.

For visualizing the expression data as heatmap and performing a hierarchical clustering of the PDX models, the packages gplots[41] (function heatmap.2) and RColorBrewer[42] were used. The dendrogram next to the (magnified) heatmaps was based on the clustering (complete linkage, Euclidean distance) of all 19096 genes.

For the investigation of gene *TP63* as marker for squamous cell carcinoma histology, the expression value cut-off 12 was set for the given data sets. PDX models with *TP63* expression values ≥ 12 were classified as squamous cell carcinoma. The obtained classification was compared to the histopathological

classification. Sensitivity, specificity, and corresponding 95% confidence intervals (CI) were calculated using package epiR[43].

Investigation for statistically significant over-representation of the selected gene set including *CXCL12, CXCR4, EGF, EGFR, HGF, KDR, MET, PTGS2,* and *VEGFA* within all HGNC-annotated genes, a single-sample gene set variation analysis (ssGSVA) combined with a permutation-based test was performed utilizing RNASeq expression data of the 38 PDX models. Considering all 19096 HGNC-annotated genes ($N$), $S \in N$ is the set of nine selected genes summarized above. In $I = 100,000$ iterations, datasets $T$ were created for which each $T$ consists of $|S|$ genes sampled from $N$. ssGSVA (implemented in R package "GSVA")[44] was performed for $S$ and $T$ based on the expression data of $N$ resulting in (scaled) enrichment scores for each set in the corresponding PDX model. The $p$-value was calculated by dividing the number of occurrences $E > F$, where $E$ represents the enrichment score of $S$ and $F$ the enrichment score of $T$, and the total number of iterations $I$.

**Mutational analysis**. The high sequencing depth of the RNASeq we performed (80–100 Million reads), allowed a sensitive mutational analysis of the transcriptome including a detection of fusion events. The variant calling and annotation of mapped reads was performed using the GATK tools version 4.0.2.1 and the Ensemble Variant Effect Predictor (VEP). For the analysis of gene fusion events the STAR-fusion tool version 1.4.0 was used.

**Analysis of oncogenic mutations utilizing a targeted exome cancer gene panel**. Total DNA from xenograft tissue samples was isolated using a commercially available column-based purification kit from QIAGEN (DNeasy Blood and Tissue Kit) according to manufacturer's instructions. Purified DNA samples were analyzed with the Illumina TruSeq® Amplicon Cancer Panel (TSACP) which targets 212 amplicons from 48 tumor-related genes in a multiplexed reaction. All necessary reagents are supplied in the TSACP kit (cat. FC-130-1008) and the amplicon library preparation was performed according to the manufacturers protocol. Finally, next-generation sequencing was carried out on an Illumina MiSeq® Desktop Sequencer. Illumina MiSeq Reporter including Somatic Variant Caller 3.1.10.0 and Human GRCh37 hg19 (Feb 2009) as reference genome was used for sequence variant calling. The annotation and interpretation of detected variants was performed using the public IGV genome browser (Integrative Genomics Viewer, Broad Institute, Cambridge MA, USA)[45]. For correlation studies only sequence variations, which passed the Illumina calling quality criteria and which occur with a frequency > 10% were considered.

**Patient-derived xenograft (PDX) lung cancer models**. For the establishment and characterization of PDX mouse models and lung PDX models, in particular, fresh tumor samples were cut into pieces of 3 × 3 mm and were subcutaneously transplanted into 6–8 weeks old *Mus musculus*, Rj:NMRI-Foxn1nu/nu female nude recipient mice[46,47]. Once subcutaneous tumors became palpable, tumor size was measured twice weekly with a digital caliper. Individual tumor volumes were calculated by the formula $V = (\text{length} \times \text{width}^2)/2$ and related to the values at the first day of treatment (relative tumor volume, RTV). In addition, therapeutic efficacy was assessed by applying Response Evaluation Criteria In Solid Tumors (RECIST)[21]. According to these criteria, a reduction in tumor volume of at least 30% was defined as partial response (PR), an increase in tumor volume of at least 20% was defined as progressive disease (PD). Stable disease (SD) was defined as neither sufficient reduction to qualify for PR nor sufficient increase to qualify for PD. The Objective Response Rate (ORR) includes all tumors with Complete Response (CR) and PR and the Clinical Benefit Rate includes all tumors with CR, PR and SD expressed in percentage. The body weight of mice was determined on a regular basis and the change in body weight was taken as variable for tolerability. After 25–40 days, the mean tumor volume reached the indicated starting volume (150–200 mm³). Mice were randomly assigned to control and treatment groups (1–4 mice per group) and treatment was started on the day of randomization.

The mice were treated using the following drug dosages and treatment schedules: cabozantinib 15 mg/kg, afatinib 15 mg/kg, etoricoxib 10 mg/kg orally and plerixafor 5 mg/kg intraperitoneally for 5 days on treatment and 2 days off treatment for four cycles in total (Table 1). Placebo control mice were treated with corresponding vehicles only (p.o. with 0.5% methyl cellulose and s.c. with aqua ad iniectabilia) or with respective monotherapy at the same dosages. At the end of the experiments, tumors were excised and one half snap frozen and stored at −80 °C for further analyses. The second half was fixed in formalin fixed and embedded in paraffin (FFPE).

All mice were handled in accordance with the Guidelines for the Welfare and Use of Animals in Cancer Research[48] and according to the German Animal Protection Law, approved by the responsible local authorities. Generation of the PDX has been approved by the local ethics committee at Charité Berlin and patients have provided informed consent.

**Histology**. For immunohistochemical staining, 3 µm thick serial sections of experimental NSCLC tissues were cut from FFPE tissue samples.

PD-L1 staining was carried out with a Bondmax automated slide staining system using the Polymer Refine Detection Kit (both Leica Biosystems, Wetzlar, Germany) and a rabbit monoclonal anti-PD-L1 antibody (1:100, clone E1L3N, Cell

Signaling, Frankfurt a.M., Germany). For staining of pan-cytokeratin, alpha-smooth muscle actin (α-SMA), and CD31, antigen retrieval was performed in a steamer for 20 min in citrate buffer pH 6.0. Blocking was carried out in PBS supplemented with 4% BSA for 1 h at room temperature (RT). Incubation with primary antibodies: mouse anti-human pan-Cytokeratin (1:50; clone A1/A3; Dako, Hamburg, Germany), rat anti-mouse CD31 (1:200; clone SZ31, Dianova, Hamburg, Germany) and mouse anti-mouse α–SMA (1:400; clone 1A4; Sigma-Aldrich Chemie Taufkirchen, Germany) was conducted 45 min at RT. Detection of primary antibodies (pan-Cytokeratin and α-SMA) was performed by using EnVision-mouse HRP (Dako, Hamburg, Germany) for 45 min at RT. Detection of anti-CD31 antibody was performed using a goat-anti-rat-HRP antibody (Dianova, Hamburg, Germany). Visualization was done by using the AEC Substrate Kit (Abcam, Berlin, Germany) for 10–20 min. After final washing in PBS, sections were stained in Mayer's Hemalaun (AppliChem, Darmstadt, Germany) for 2 min. After washing in water for 10 min, cover slips were fixed with Kaiser's glycerin gelatine (Waldeck, Munster, Germany). Specificity of all stainings has been ensured before by application of a respective isotype control revealing no or only weak staining. All evaluations were performed twice in a blinded manner.

**Statistics and reproducibility**. The number of independent biological replicates of PDX-tumor bearing mice used in every experiment is specified in the corresponding figure legend. Information on bioinformatics statistics and data processing are described in the specific methods section.

**Reporting summary**. Further information on research design is available in the Nature Research Reporting Summary linked to this article.

## Data availability

Source data for the graphs and charts in the main figures are provided in Supplementary Data 1. All data on the characterization and the DNA sequencing of the PDX models are available from the EPO tumor biobank at http://www.epo-berlin.com/epo-tumor-models-xenografts.html. Access will be granted by sending an email to service@epo-berlin.com. Any remaining information can be obtained from the corresponding author upon reasonable request.

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

## Acknowledgements

We gratefully thank Lars Haakon Søraas from Norway who has inspired the reuptake of this research. We thank Svetlana Gromova and Britta Büttner for technical assistance for the PDX in vivo studies and Sandra Ussat and Maike Witt-Ramdohr for the immunohistochemical stainings.

## Author contributions

D.G., T.C., M.B., S.S, and C.R. conducted experiments and generated data. D.G., T.C., M.B., S.S, C.R., J.H., and S.L. analyzed and interpreted data. S.L., J.H., and D.G. conceived, designed, and/or supervised experiments. S.L., D.G., J.H., and S.S. wrote the main manuscript. All authors approved the final manuscript.

## Competing interests

J.H., D.G., and S.L. have issued a related patent, PCT/EP2021/072486. J.H. declares the employment and the ownership of the EPO GmbH. The remaining authors declare no competing interests.
