## [Peer Review File · Communications Biology]

Reviewers' comments:

Reviewer #1 (Remarks to the Author):

The manuscript reports on the preclinical evaluation of the efficacy of a novel combination of different agents in NSCLC patient-derived xenografts.

Some comments:

- The sentence in the Discussion "Despite some recent progress in the treatment of NSCLC, no impactful advances have been made in tumors without an actionable driver mutation and without the expression of PD-L1 >50%" is not longer completely true (ref. 4 is dated back to 2018) and must be revised. The use of chemo-immunotherapy combinations has recently dramatically changed the therapeutic scenario of 1st line therapy in advanced/metastatic NSCLC regardless of PD-L1 expression [Borghaei H, et al. Cancer 2020; Gandhi L, et al. NEJM 2018; Paz-Ares L, et al. NEJM 2018; Paz-Ares L, et al. J Thorac Oncol 2020]. The survival results obtained with these combinations is by far superior to that obtained with chemotherapy alone and represents a major step forward in the therapeutic management of non-oncogene-addicted NSCLC. Please rephrase and use a more recent reference(s).
- The major concern is the toxicity of this regimen. In the Discussion the authors briefly discuss this issue. However, it should be mentioned that some of the toxicities observed with these agents when used singularly might coincide when used simultaneously. For instance, the combination afatinib-cetuximab when used in clinics reported high grade of skin and gastrointestinal toxicities making this promising combination less appealing compared with other less toxic therapeutic options (Janjigian YY, et al. Cancer Discov 2014; Goldberg SB, et al. J Clin Oncol. 2020). Furthermore, no details on potential pharmacological interactions have been discussed/included.
- Figure 2d reports on the patient demographics, tumor characteristics and best response per RECIST in patient-derived xenografts. No driver mutations were identified. The definition of non-oncogene addicted NSCLC is changing rapidly as novel oncogenic drivers are increasingly added to the list of exploitable targets in this disease. It would be useful to provide the complete list of genes analyzed in these tumors, as some of the targeted agents used here might block unusual oncogene drivers that might not be tested (but could be actually present). For instance, cabozantinib is also a RET inhibitor and RET rearrangements are reported in a very minor fraction of NSCLC (Li AY, et al. Cancer Treatment Reviews 2019) and afatinib has been reported to target the rare NRG1 rearrangement (Gay ND, et al. J Thorac Oncol 2017)
- The manuscript contains some typos and a linguistic revision might be useful in a few sentences

Reviewer #2 (Remarks to the Author):

In the article titled "Breaking the crosstalk of the cellular tumorigenic network in NSCLC by a highly effective drug combination", authors aimed to overcome the treatment failure or resistance in a subclass of NSCLC tumors which don't have any oncogenic driver mutations. Author hypothesized that combination of four drugs treatment which targets multiple pathways might overcome resistance. Unfortunately, the expression profile of nine genes involved in multiple pathways used for targeting were not clustered exclusively among treatment failure progressive disease (PD)PDXs (Fig 2b). Moreover, only 8 out 38 PDXs had significantly higher expression of these 9 genes. 12 out of 38 PDXs showed PD, where 5 of them significantly upregulated these nine genes and 7 of them are PD but not associated with nine gene expression. Although, the four drug combination tested on selected 5 PDXs (two are PD, three are SD or PR) showed antitumor activity regardless of gene expression status (nine genes expression) or disease progression status. The antitumor activity showed during treatment period. PDX tumors seem like growing back after treatment indicating there might be other pathways involved or evolved.

Although the four drug combination showed effectiveness to these subtypes of NSCLC, but the evidence doesn't clearly support the hypothesis. Authors could do more analysis (such as gene expression, pathway) of these 38 PDXs especially PD PDXs and find out the active pathway (s) for treatment failure and overcome the resistance by blocking the active pathways. Also, author could do the analysis of residual PDX tumors (after treatment) to determine the bypass mechanism for acquired resistance of this combination.

Dr. Stefan Langhammer
life science consulting
Hirschweg 8
30938 Burgwedel
Germany

mobile phone: +49 171 411 9744
email: langhammer@ls-consultant.net

Burgwedel, November 15th 2021

COMMSBIO-21-2043-A

Revision: Breaking the Crosstalk of the Cellular Tumorigenic Network in NSCLC by a Highly Effective Drug Combination

Dear Sir or Madam,

Thank you very much for your thorough review of our manuscript. Your comments greatly helped to clarify and to improve our research article leading to a substantial revision including the incorporation of additional data.

Please kindly find our answers to the reviewers' comments in detail below. All changes have been highlighted by underlining throughout the manuscript.

With best regards
Stefan Langhammer

Reviewer #1

Comment 1

"The sentence in the Discussion "Despite some recent progress in the treatment of NSCLC, no impactful advances have been made in tumors without a targetable oncogenic mutation and without the expression of PD-L1 >50%" is not longer completely true (ref. 4 is dated back to 2018) and must be revised. The use of chemo-immunotherapy combinations has recently dramatically changed the therapeutic scenario of 1st line therapy in advanced/metastatic NSCLC regardless of PD-L1 expression [Borghaei H, et al. Cancer 2020; Gandhi L, et al. NEJM 2018; Paz-Ares L, et al. NEJM 2018; Paz-Ares L, et al. J Thorac Oncol 2020]. The survival results obtained with these combinations is by far superior to that obtained with chemotherapy alone and represents a major step forward in the therapeutic management of non-oncogene-addicted NSCLC. Please rephrase and use a more recent reference(s).."

Answer to comment 1: We agree with the reviewer that the treatment options for NSCLC patients without a targetable oncogenic mutation and the expression of PD-L1 < 50% has changed significantly as reflected by the pooled analysis of 3 randomized controlled trials by Borghai et al., 2020 revealing an ORR of about 50% in this patient population.

Based on this comment we revised the manuscript as follows:

Page 2:

The respective reference was cited for the following sentence in the Introduction:” These mainly non-resectable carcinomas are treated with platinum-based chemotherapy alone or in combination with immune-checkpoint inhibitors, which are the mainstay regimens in the absence of predictive, targetable oncogenic mutations and an expression of the immune-checkpoint inhibitor Programmed death ligand-1 (PD-L1) of <50%¹.“

Page 12:

The respective sentence in the discussion section was rephrased and the reference was included as follows: “Despite some impactful advances in the treatment of NSCLC without targetable oncogenic mutations and without expression of PD-L1 resulting in an ORR of about 50%¹, the unmet medical need for many patients in this population remains high⁵.“

Reference added to the manuscript:

Borghaei H, Langer CJ, Paz-Ares L, et al. Pembrolizumab plus chemotherapy versus chemotherapy alone in patients with advanced non-small cell lung cancer without tumor PD-L1 expression: A pooled analysis of 3 randomized controlled trials. *Cancer*. 2020;126(22):4867-4877.

Comment 2

“The major concern is the toxicity of this regimen. In the Discussion the authors briefly discuss this issue. However, it should be mentioned that some of the toxicities observed with these agents when used singularly might coincide when used simultaneously. For instance, the combination afatinib-cetuximab when used in clinics reported high grade of skin and gastrointestinal toxicities making this promising combination less appealing compared with other less toxic therapeutic options (Janjigian YY, et al. *Cancer Discov* 2014; Goldberg SB, et al. *J Clin Oncol*. 2020).“

Answer to comment 2: In general, we agree with the reviewer that the toxicity and the side effect management of combination regimens may display a significant challenge in the treatment of cancer patients.

However, the combination drug regimen we compiled is a low dose regimen that, in addition to that, is applied in a lower frequency than usually used for respective monotherapies (5 days on treatment and 2 days off treatment versus once daily).

The drug dosage for cabozantinib in this regimen (15mg/kg) is less than half of the dosage normally used for *in vivo* monotherapies (30-60mg/kg). Similar to that, a 20% lower dose (15mg/kg) for afatinib is used compared to other *in vivo* monotherapies (≥ 20 mg/kg). Both compounds, cabozantinib and afatinib have the most pronounced safety profile in the clinical setting when compared to etoricoxib and plerixafor.

In addition, in this combination drug regimen we are not addressing the same pathway mechanism twice as it would be the case for the combination of afatinib plus cetuximab which may lead to an increase of on-target related toxicity.

Thus, based on our results for the toxicity analysis we would assume that an equivalent extrapolated low dose drug regimen in the clinical setting would have a manageable safety profile.

To make this point clear to the readership we added table 1 on Page 6 highlighting the characteristics of this low dose regimen in comparison to common dosing and treatment schedules for respective monotherapies.

Page 10:

The following chapter in the Results section is intended to describe the toxicity results of our experiment, highlighting the manageable safety profile in the treated mice:

“In none of the remaining mice at any time point a significant difference in BW between the verum and the placebo or the monotherapy group was observed. In general, all mice included in these experiments either gained weight during the course of treatment or remained at the baseline BW level with minimal variances (Fig. 3). None of the mice had to be discontinued from treatment, died or had to be euthanized during the course of treatment. This observation reflects the beneficial safety profile of this combination regimen and is in alignment with the low dose therapy compared to regular concentrations used of these drugs in *in vivo* experiments (Table 1).”

Page 12:

The following existing chapter in the Discussion section is reflecting on the potential clinical safety profile of an equivalent low dose combination regimen:

“The combination therapy regimen that was compiled on the five criteria published previously⁷ can be considered as a low dose treatment regimen. The two drugs with a more pronounced safety profile in this therapy regimen are cabozantinib and afatinib. Cabozantinib is approved for the treatment of renal cell carcinoma and hepatocellular carcinoma and is efficacious in different tumor models at dosages between 30-60mg/kg once a day^{43,36,44}. Therefore, the low dose drug combination use in our study only includes 50% or less of typically used cabozantinib concentrations in recently published preclinical studies. Afatinib is approved for the treatment of NSCLC either with activating EGFR mutation or in squamous cell carcinoma under or after progression on platinum-based chemotherapy and is efficacious in tumor models at dosages from 20mg/kg or higher once a day³⁹. In the combination therapy regimen

used in this preclinical study, both drugs were used significantly below these drug doses, each at 15mg/kg and in addition to that paused after every 5 days of treatment. Plerixafor, approved for stem cell mobilization, and etoricoxib, approved for pain treatment and degenerative and auto-inflammatory diseases, were used at a commonly used concentration levels in *in vivo* models^{38,40}. The usage of this combination regimen showed an excellent safety profile and might lead to less pronounced side effects in patients. Together with its excellent efficacy this combination would be superior to a higher prescribed medication of single cabozantinib or afatinib.”

Comment 3

“Furthermore, no details on potential pharmacological interactions have been discussed/included.”

Answer to comment 3: We also agree with the Reviewer that a drug-drug interaction (DDI) analysis would improve the information on the compiled combination regimen. Therefore we have run an online DDI analysis at https://www.drugs.com/drug_interactions.html. The only possible DDI interaction that was found is a moderate interaction between afatinib and cabozantinb due to a potential increase of afatinib plasma concentrations by P-glycoprotein (P-gp) inhibitors. This interaction only was found when the P-gp inhibitor ritonavir was administered one hour before afatinib but not when it was administered simultaneously. The recommendation for the combined usage of these drugs is a close monitoring of the patients which should be possible in the setting of a clinical trial. Due to unavailability in this DDI analysis etoricoxib was replaced with the celecoxib from the same drug class.

Based on this comment we revised the manuscript as follows:

Page 13:

The following chapter in the Discussion section was added:

“A database analysis for potential drug-drug interactions of this drug combination only revealed a potentially moderate clinically significant interaction between cabozantinib and afatinib that may result in a plasma level increase of afatinb when administered after exposure to a P-glycoprotein (P-gp) inhibitor. A monitoring of patients for the onset of adverse effects is recommended. Due to unavailability in this DDI database etoricoxib was replaced with celecoxib, another drug of the identical drug class of selective COX-2 inhibitors⁴⁸”

Reference added to the manuscript:

Drugs.com [Internet]. Drug Interactions Checker: afatinib, cabozantinib, celecoxib, plerixafor; c2000-2021 [Updated: 14 October 2021, Cited: 17 October 2021]. Available from: https://www.drugs.com/drug_interactions.html.

Comment 4

“Figure 2d reports on the patient demographics, tumor characteristics and best response per RECIST in patient-derived xenografts. No driver mutations were identified. The definition of non-oncogene addicted NSCLC is changing rapidly as novel oncogenic drivers are increasingly added to the list of exploitable targets in this disease. It would be useful to provide the complete list of genes analyzed in these tumors, as some of the targeted agents used here might block unusual oncogene drivers that might not be tested (but could be actually present). For instance, cabozantinib is also a RET inhibitor and RET rearrangements are reported in a very minor fraction of NSCLC (Li AY, et al. Cancer Treatment Reviews 2019) and afatinib has been reported to target the rare NRG1 rearrangement (Gay ND, et al. J Thorac Oncol 2017)

Answer to comment 4:

We thank the reviewer for this important comment allowing to clarify our strategy. For our experiments we selected tumors without an identified targetable oncogenic mutation from the PDX panel only.

Therefore, we utilized the Illumina MiSeq Exome Cancer Panel evaluating 212 amplicons from 48 tumor related genes (*ABL1, EGFR, GNAS, MLH1, RET, AKT1, ERBB2, HNF1A, MPL, SMAD4, ALK, ERBB4, HRAS, NOTCH1, SMARCB1, APC, FBXW7, IDH1, NPM1, SMO, ATM, FGFR1, JAK2, NRAS, SRC, BRAF, FGFR2, JAK3, PDGFRA, STK11, CDH1, FGFR3, KDR, PIK3CA, TP53, CDKN2A, FLT3, KIT, PTEN, VHL, CSF1R, GNA11, KRAS, PTPN11, CTNNB1, GNAQ, MET, RB1*).

The existence of gene rearrangements for *ALK, ROS, RET* and *NTRK* was assessed by a highly sensitive RNASeq mutational analysis.

Based on this comment we revised the manuscript as follows:

Page 5 and page 6:

The following chapters in the Methods section were added:

“Mutational analysis

The high sequencing depth of the RNASeq we performed (80-100 Million reads), allowed a sensitive mutational analysis of the transcriptome including a detection of fusion events. The variant calling and annotation of mapped reads was performed using the GATK tools version 4.0.2.1 and the Ensemble Variant Effect Predictor (VEP). For the analysis of gene fusion events the STAR-fusion tool version 1.4.0 was used.

Analysis of oncogenic mutations utilizing a targeted exome cancer gene panel

Total DNA from xenograft tissue samples was isolated using a commercially available column-based purification kit from QIAGEN (DNeasy Blood and Tissue Kit) according to manufacturer’s instructions. Purified DNA samples were analyzed with the Illumina

TruSeq® Amplicon Cancer Panel (TSACP) which targets 212 amplicons from 48 tumor related genes in a multiplexed reaction. All necessary reagents are supplied in the TSACP kit (cat. FC-130-1008) and the amplicon library preparation was performed according to the manufacturers protocol. Finally, next generation sequencing was carried out on an Illumina MiSeq® Desktop Sequencer. Illumina MiSeq Reporter including Somatic Variant Caller 3.1.10.0 and Human GRCh37 hg19 (Feb 2009) as reference genome was used for sequence variant calling. The annotation and interpretation of detected variants was performed using the public IGV genome browser (Integrative Genomics Viewer, Broad Institute, Cambridge MA, USA)³³. For correlation studies only sequence variations, which passed the Illumina calling quality criteria and which occur with a frequency >10 % were considered.”

Page 9 and page 10:

The following chapter in the Results section was added:

“The existence of targetable oncogenic mutations was excluded by utilizing the Illumina amplicon cancer panel (TSACP), sequencing 212 regions in 48 cancer related genes including *EGFR*, *BRAF*, *KRAS*, *MET* and *RET* (Fig. 2d). As an example, in this panel the *KRAS G12C* mutation was identified in the highly treatment resistant PDX tumor Lu7198 and therefore this model was excluded from further experiments at this stage. In none of the selected PDX models a gene rearrangement of *ALK*, *ROS*, *RET* or *NTRK* was identified as assessed by RNASeq mutational analysis.”

Figure 2d.: In order to display the identified mutations from the Illumina Cancer Gene Panel, the column “Cancer Gene Panel Mutation Analysis” was added, showing all identified mutations occurring with a frequency of >10% within this panel. By thorough literature search none of these mutations was identified as a targetable oncogenic mutation in NSCLC.

Reference added to the manuscript:

Robinson JT, *et al.* Integrative genomics viewer. *Nat Biotechnol* **29**, 24-26 (2011).

Comment 5

“The manuscript contains some typos and a linguistic revision might be useful in a few sentences”

Answer to comment 5: We have spell-checked the manuscript and a linguistic check was performed by a native speaking colleague. We have added some minor changes in this respect.

Reviewer #2

Comment 1:

“Unfortunately, the expression profile of nine genes involved in multiple pathways used for targeting were not clustered exclusively among treatment failure progressive disease (PD)PDXs (Fig 2b). Moreover, only 8 out of 38 PDXs had significantly higher expression of these 9 genes. 12 out of 38 PDXs showed PD, where 5 of them significantly upregulated these nine genes and 7 of them are PD but not associated with nine gene expression. Although, the four drug combination tested on selected 5 PDXs (two are PD, three are SD or PR) showed antitumor activity regardless of gene expression status (nine genes expression) or disease progression status. The antitumor activity showed during treatment period. PDX tumors seem like growing back after treatment indicating there might be other pathways involved or evolved.

Answer to comment 1: We thank the reviewer for this comment. With the aid of the ssGSVA analysis we were able to estimate the importance of our selected nine genes within the tumorigenic network. The obtained high enrichment scores for all 38 PDX models show an over-representation of the selected gene set within the considered set of 19096 HGNC annotated genes. In addition, in 8 out of 38 models, this gene set is significantly higher enriched compared to randomly generated gene sets. The majority of these models (five out of eight) are classified as highly resistant tumors with PD which is indicative of a potential relationship between the significant over-representation of the nine selected genes and the responsivity of PDX models. However, we agree with the reviewer that no final statement can be made regarding this relationship as there are also 7 out of 30 PDX models that are associated with PD but do not show a significant over-representation of the selected gene set. Viewed in this light, our results do not provide evidence, but rather show a trend on a biological background that indicates a possible correlation.

Based on this comment we revised the manuscript as follows:

Page 12:

“We provide evidence that the expression of a respective set of target genes is important for NSCLC development, homeostasis and progression. Our ssGSVA obtained enrichment scores indicate the relevance of the underlying biological function of the selected pathways in NSCLC disease and treatment. Observing a relatively low statistically significant over-representation of the nine target genes for only 8 of 38 models by permutation-based test is even more confirmative highlighting the great range of individual NSCLC characteristics and phenotypes rather than pointing out a bias with our analysis.”

Comment 3

“Although the four drug combination showed effectiveness to these subtypes of NSCLC, but the evidence doesn't clearly support the hypothesis. Authors could do more analysis (such as gene expression, pathway) of these 38 PDXs especially PD PDXs and find out the active pathway (s) for treatment failure and overcome the resistance by blocking the active pathways.

Answer to comment 3: We agree with the reviewer that an additional characterization of the PDX panel for potential resistance mechanisms would be helpful for the overall understanding of the resistance profile, as displayed as “Resistance Profile” based on RECIST responses in Fig. 3a. We assume that a variety of mechanisms lead to the resistance for the respective tumors against the drugs tested here.

In a significant number of tumors from this PDX panel, including all tumors reported in this article, the resistance mechanism against carboplatin was analyzed uncovering *LRP12* DNA methylation as a predictive biomarker for carboplatin resistance (Grasse et al., 2018, e.g. Figure 4c).

Based on this comment we revised the manuscript as follows:

Page 10:

The following sentence and reference was added to the Results section:

“All selected tumors from the NSCLC PDX panel were previously analyzed and reported for the identification of *LRP12* DNA methylation as a predictive biomarker for carboplatin resistance (Grasse et al., 2018).”

Reference added to the manuscript:

Grasse S, Lienhard M, Frese S, et al. Epigenomic profiling of non-small cell lung cancer xenografts uncover *LRP12* DNA methylation as predictive biomarker for carboplatin resistance. *Genome Med.* 2018;10(1):55.

Comment 4

Also, author could do the analysis of residual PDX tumors (after treatment) to determine the bypass mechanism for acquired resistance of this combination.”

Answer to comment 4: We agree with the reviewer that the mechanisms for acquired resistances against targeted therapy regimens are of particular importance. However, in our experiments no acquired resistance against this combination therapy regimen was induced at any point of time. Only a slow re-growth of the tumors was observed several days after treatment discontinuation reflecting a lack of treatment pressure on the tumors. Therefore, we would not be able to identify pathways that may correlate with any kind of resistance against this combination regimen in our experiments.

For further improvement, the following additional changes were made:

- Sample sizes and definition of error bars were added to the figure legends
- The title of Fig.3 was rephrased from “Tumor volume and body weight” to “Efficacy of the combination therapy regimen in NSCLC PDX mouse models”
- The phrase “identified driver mutation” was replaced by “targetable oncogenic mutation” throughout the manuscript

REVIEWERS' COMMENTS:

Reviewer #1 (Remarks to the Author):

The authors have addressed all the issues raised during the first revision round. I have no further comments/suggestions

Reviewer #2 (Remarks to the Author):

Authors satisfactorily answered most of the comments/questions raised based on earlier version of the manuscript. It would significantly increase the quality of work, if this combination is tested against all 5 PD (progress disease) PDXs.

Dr. Stefan Langhammer
life science consulting
Hirschweg 8
30938 Burgwedel
Germany

mobile phone: +49 171 411 9744
email: langhammer@ls-consultant.net

Burgwedel, December 12th 2021

COMMSBIO-21-2043B

Final revision: Breaking the crosstalk of the Cellular Tumorigenic Network by low-dose combination therapy in NSCLC patient-derived xenografts

Dear Sir or Madam,

Again, thank you very much for your thorough review of our manuscript!

Please kindly find our answer to your comment below. All changes have been highlighted by underlining throughout the manuscript.

With best regards
Stefan Langhammer

Reviewer #2

Comment 1

“It would significantly increase the quality of work, if this combination is tested against all 5 PD (progress disease) PDXs.”

Answer to comment 1: We agree with the reviewer that a further validation of this combination therapy should be carried out in additional tumor models and eventually in different tumor indications.

In order to clarify this aspect, we revised the manuscript in the final sentence of the discussion section as follows:

Page 10: “Based on our results we encourage the further validation of this combination therapy in additional cancer models and we recommend the setup of a clinical study...”